# Co-design of Lifestyle6, a digital tool targeting multiple health behaviour changes for cancer risk reduction and early detection support

Beatrice Murawski[1,2], Katelyn Emma Collins[1,3]*, Bianca Viljoen[1,4], Emma Charlotte Pearse[1], Jazmin Vicario[1], Anna Stiller[1], Hannah Grennan[5], Grant Brown[5], Lee Woods[6], Sonja March[7], Belinda Goodwin[1,7,8]

**1** Viertel Cancer Research Centre, Cancer Council Queensland, Fortitude Valley, Queensland, Australia, **2** School of Public Health, University of Queensland, Herston, Queensland, Australia, **3** School of Psychology and Wellbeing, University of Southern Queensland, Springfield, Queensland, Australia, **4** School of Nursing and Midwifery, University of Southern Queensland, Toowoomba, Queensland, Australia, **5** Prevention and Early Detection, Cancer Council Queensland, Fortitude Valley, Queensland, Australia, **6** Queensland Digital Health Centre, Faculty of Health, Medicine and Behavioural Sciences, The University of Queensland, Herston, Queensland, Australia, **7** Centre for Health Research, University of Southern Queensland, Springfield, Queensland, Australia, **8** School of Global and Population Health, University of Melbourne, Victoria, Australia

\* u1160643@umail.usq.edu.au

## Abstract

Up to 40% of all cancers diagnosed could be prevented through the uptake of healthy lifestyle behaviours. This paper describes the co-design of a novel community-centric digital programme for cancer prevention and early detection support. Twenty-two community panel members partook in a series (i.e., three group-based sessions followed by one individual session) of iterative online workshops to co-design the programme. Based on established Design Thinking principles, the workshops aimed to i) identify barriers to access and use of existing cancer prevention information and support, ii) explore consumers' needs and preferences for information and support related to cancer risk reduction and early detection, iii) generate ideas for a digital solution to address those, and iv) seek panel members' feedback on a digital prototype developed in line with the insights from previous stages. The key barriers to accessing cancer prevention and screening support identified by the panel included limited availability and knowledge of cancer-specific resources alongside a paradoxical information overload regarding generic options. Panel members expressed concerns about information accuracy and relevance, while time constraints, financial limitations, and motivation deficits further impeded resource engagement. Regarding prototype preferences, participants prioritised accessibility, diversity, communication and connectedness, usability, and personal relevance as essential design elements. Digital solutions for cancer prevention and screening support should offer a customised experience, while catering for varying practical constraints and motivational challenges.

**Data availability statement:** The data that comprise the findings of this study are held securely on password protected servers and are only currently accessible by members of the research team granted permission from the University of Southern Queensland Human Research Ethics Committee (ref. ETH2023-0359). The data are unable to be uploaded to a publicly available server due to the potentially identifying nature of the information; however, an aggregated summary of the data may be requested by other parties (e.g., researchers) upon reasonable request to the Research Operations Team at the Viertel Cancer Research Centre (research@cancerqld.org.au). The merit of each request will be individually evaluated before provision of de-identified interview materials.

**Funding:** This research was funded by Cancer Council Queensland. The funders had no role in study design, data collection and analysis, decision to publish, or preparation of the manuscript.

**Competing interests:** The authors have declared that no competing interests exist.

## Introduction

With approximately 20 million new cases and 10 million deaths per year, [1] cancer remains a significant global health challenge, placing a significant burden on healthcare systems worldwide. While cancers often arise from immutable factors such as DNA damage and genetic risk factors, a host of lifestyle factors such as tobacco and alcohol use, physical inactivity, and dietary behaviours are also associated with elevated cancer risk [2]. In fact, research indicates that 30–50% of cancers (particularly solid tumours) could be prevented through the adoption of healthy lifestyle behaviours [2–4,5]. As such, encouraging and empowering members of the public to consume a balanced diet, engage in regular physical activity, avoid tobacco products and alcohol, and use sun protection could drastically reduce the global cancer burden [6,7]. In addition, among cancers which can be reliably screened for (e.g., breast, bowel, and cervical), screening programs can significantly enhance early detection, reducing mortality rates [8]. However, such screening programs do not exist for most cancers, and are typically only available for subsets of the population identified to be at elevated risk [9]. Therefore, it is critical to consider lifestyle interventions to reduce cancer risk which may benefit the population as a whole.

Evidence suggests that "spillover" effects are observed between various modifiable risk factors, wherein improvements in one lifestyle behaviour often promote improvement in others [10]. For example, increased physical activity is known to promote better dietary choices and enhance sleep patterns, with some behaviours (e.g., physical activity and sleep) showing a bidirectional relationship [11,12]. Despite their common co-occurrence and reciprocity [13–15], interventions aimed at improving lifestyle behaviours typically target single risk factors in isolation. This may represent a missed opportunity, given those with multiple risk factors have a higher chance of getting cancer and also face susceptibility to a wider range of cancer types than those with one or no risk factors [16]. However, complex behaviour change programmes that deliberately leverage the interconnectedness between behaviours are required to maximise outcomes.

Mobile health (mHealth) solutions offer distinct advantages to meet the requirements of complex behaviour change programmes where broad reach, high-level tailoring and extended support are critical [17]. They are relatively easy to disseminate, scale up and adapt, whilst enabling sophisticated personalisation through adaptive algorithms that respond to individual user characteristics, preferences, and progress to deliver well-calibrated interventions when needed [17,18]. The persistent presence of smartphones in daily life creates natural opportunities for gentle nudges that can reinforce behaviour change principles without requiring substantial time commitments. Furthermore, mobile platforms can incorporate interactive features such as goal-setting tools, progress visualisation, and achievement systems that support the complex psychological mechanisms and individual barriers underlying behaviour change [19].

Recent reviews of mHealth behaviour change interventions indicate outcomes are mixed, with intervention effects typically ranging from small to moderate [20–32], with low user engagement emerging as a primary limiting factor [33]. The reasons for poor

engagement include a lack of perceived relevance, insufficient personalisation, limited usability, and individual personal barriers [34,35]. A fundamental issue underlying these challenges is that most programmes are designed without meaningful end-user involvement, with user feedback often restricted to post-implementation evaluation.

The use of co-design approaches such as Design Thinking in intervention development have emerged in attempt to combat this issue [36–38]. Design Thinking aims to engage consumers in the conceptualisation, development and refinement of interventions, ensuring that the end product is representative of the target population and their lived experiences [34,35]. This, in turn, may increase the likelihood of successful programme uptake and sufficient engagement [39]. Moreover, this collaborative approach ensures that health promotion initiatives are both socially acceptable and culturally appropriate [40,41], thereby maximising the potential reach and impact of the intervention.

To date, a cancer-specific behaviour change programme with the capacity to address multiple health behaviours has yet to be developed, and the needs and preferences of consumers likely to engage in such a platform are unknown. Such targeted approaches could better address the contextual challenges of preventing particular diseases (e.g., cancer) and potentially yield better engagement rates [42,43], with the added benefit that risk factors known to reduce cancer risk also reduce the risk of other non-communicable diseases [44]. This manuscript describes the design (i.e., Phase I) of *Lifestyle 6* – a novel digital cancer risk reduction and early detection support programme that was co-designed with a large panel of community members.

## Methods

### Study design

This qualitative study was informed by the Learning Health Systems Framework [44] and the Behaviour Change Wheel [45]. The stage-wise community consultation process detailed here, followed established Design Thinking principles [36]. Design Thinking is a powerful approach to human-centered digital design that offers a multi-stage method for the iterative design and testing of user-informed solutions [42]. As such, it suits the complex challenges associated with cancer risk reduction and early detection. The stages of Design Thinking include: *Empathise* (aiming to understand the issue as experienced by those affected), *Define* (in which users generate guiding problem statements to anchor the design process), *Ideate* (in which users generate radical design alternatives), and *Prototype* (in which user input is synthesised into a trial version of the final product or system presented to users for feedback) [42]. throughout these stages, the Behaviour Change Wheel was integrated to facilitate the systematic translation of qualitative insights into evidence-based, actionable suggestions to inform development of a program capable of eliciting lifestyle behaviour change [45]. This manuscript follows the consolidated criteria for reporting qualitative research (COREQ) (see S1 File) [46,47], a 32-item checklist describing best practice reporting of the methodology, design, and analysis of qualitative interview and focus group studies The study received ethical approval from the University of Southern Queensland Human Research Ethics Committee (Reference #: ETH2023−0359).

### Setting

Cancer Council Queensland is a not-for-profit organisation dedicated to reducing the impact of cancer in the Australian State of Queensland. The organisation occupies a unique position to drive public awareness related to cancer risk and early detection, provide useful resources, and deliver behaviour change support. To enhance innovation and ensure diversity of perspective, a group of subject matter experts in behavioural science, health psychology, epidemiology, service delivery, digital health, human-centred design and eLearning was actively involved in the various project stages. The project team is described in detail in S2 File.

### Participants and recruitment

Participants were eligible if they were English-speaking Queensland residents, aged 18 or older, had internet access on a device that supports videoconferencing, and availability to attend four online panel meetings. Participants (herein referred

to as community panel members) were purposively recruited through a paid, targeted social media (i.e., Facebook) campaign throughout November 2023. Social media advertisements included a link to an electronic expression of interest (EOI) form (see S3 File) hosted on Research Electronic Data Capture (REDCap). Interested individuals were screened against pre-defined eligibility criteria and asked to provide their main reason for participation, occupation, ethnicity and availability (i.e., mornings, afternoon, evenings). This information was used to facilitate member selection and group allocation, aiming for heterogeneity between groups and homogeneity within groups. Selected individuals were notified via email and asked to provide informed consent to participate (including consent to publish de-identified excerpts of the interview transcripts) via reply email. Eligible individuals also received a copy of the full participant information statement (S4 File), a privacy and confidentiality agreement (S5 File) and a set of ground rules (S6 File) for participation in the online workshops. After obtaining consent from the selected individuals, the community panel was divided into five smaller groups. After each attendance, panel members were reimbursed with grocery eGift vouchers valued at $50 per hour (i.e., $100 per 2-hour workshop).

**Data collection**

The design process used in this study encompassed three rounds of group-based workshops (i.e., design stages *Empathise, Define,* and *Ideate*), and one round of individual workshops (i.e., design stage *Prototype*). All workshops were held online. A minimum of two and no more than eight panel members were invited to each group workshop. Workshop structures and activities were standardised for all sessions. All facilitators were briefed prior to the workshops commencing and followed a set, pilot-tested session script and protocol.

The group-based workshop activities completed in the first three stages generally encouraged divergent thinking without any major design limits imposed, in alignment with Design Thinking Principles [42]. In contrast, the Stage 4 (*Prototype*) workshops were conducted in one-on-one format and took a convergent approach, where the goal was to integrate the input from previous workshops to arrive at a solution that reflects collective understanding and agreement. A *Think Aloud* protocol was used for these sessions to encourage panel members to verbalise their thoughts and viewpoints as they perused the prototype [48]. Further details on the format, duration, activities, data collection and analyses by each Design Thinking stage is provided in Table 1. During the workshops, the digital canvas platform, Padlet, was used [49]. This allowed panel members to collaboratively brainstorm, sharing ideas and links to other media (such as Word documents) in real time. Upon completion of all focus groups, panel members were invited to complete the Public and Patient Engagement Evaluation Tool (PPEET) 2.0 [50,51] to provide feedback on their experience throughout the engagement process (see S7 File).

Given the established benefits of online interview methods, including greater convenience, participant reach, and retention [52], all workshops were hosted on Microsoft Teams, recorded, and auto-transcribed. Transcripts were de-identified and cleaned by the facilitators prior to analysis. The cleaned transcripts were not returned to the panel members for comment or correction as the session recordings were used to validate transcripts where clarification was required. Field notes were taken during the online workshops and consolidated afterwards. S8 File contains further details on session structure and content.

Online workshops were scheduled in a sequential pattern, where all workshops per stage were completed, and the associated data cleaned, consolidated and synthesised prior to the next stage commencing. Insights from the between-stage data syntheses were used to inform the content and direction of subsequent stages and workshops and the prototype as such. Where summaries of the preceding stage were presented (i.e., Stages 2–4), panel members also had the opportunity to comment on the accuracy of key insights. This approach helped maintain continuity between workshops and alignment with key project goals and desired outcomes. It further enhanced engagement due to panel members feeling validated and valued, which fostered a sense of ownership and impact, demonstrating transparency regarding the decision-making process [53].

**Table 1. Overview of panel workshops and key activities by Design Thinking stage.**

| Stage | Aims/Objectives | Format/Duration | Activities | Data collected | Analysis |
|---|---|---|---|---|---|
| 1. Empathise | To explore commonly experienced frustrations and barriers and how they relate to general perspectives, attitudes and/or previous experiences (e.g., usual engagement) with cancer risk reduction and screening programmes or services. | 5 groups with 3–7 panel members (N = 22); supported by two facilitators; 120 min duration | • Icebreaker<br>• Alignment<br>• Open discussion | • Microsoft Teams recordings (10h)<br>• Transcripts<br>• Chat history<br>• Padlet posts | • Thematic analysis<br>• Empathy maps<br>• Personas |
| 2. Define | To define the problem in clear terms. This involved synthesising information from the empathy stage to articulate the key frustrations and barriers that need addressing and revisit those with panel members to form an actionable problem statement and a corresponding needs statement. | 4 groups with 2–8 panel members (n = 21); supported by two facilitators; 120 min duration | • *Empathise* session recap<br>• Review personas<br>• Define problems and needs<br>• Open discussion | • Microsoft Teams recordings (8h)<br>• Transcripts<br>• Chat history<br>• Padlet posts | • Thematic analysis<br>• Problem statement<br>• Needs statement |
| 3. Ideate | To brainstorm a wide range of ideas and potential solutions to address the previously defined problems and barriers, followed by ranking of ideas and solutions proposed by end users based on priority or perceived importance to help identify core components or features that are considered non-negotiable and should form part of the initial prototype/s. | 4 groups with 2–6 panel members (N = 13); supported by two facilitators; 120 min duration | • *Define* session recap<br>• Brainstorming<br>• Open discussion | • Microsoft Teams recordings (8h)<br>• Transcripts<br>• Chat history<br>• Padlet posts | • Thematic analysis |
| 4. Prototype | To create a low-fidelity representation of the ideas generated in the previous stage to visualise key concepts and gather user feedback (i.e., strengths vs. areas for improvement) for further refinement using a Think Aloud protocol. | N = 17 individual (one-on-one) workshops; each supported by one facilitator; 60 min duration | • *Ideate* session recap<br>• Think Aloud protocol | • Microsoft Teams recordings (17h)<br>• Transcripts | • Thematic analysis |

*Note.* Stage 5 (i.e., to implement and test the program) will involve testing and evaluating the final platform by real-world users to determine reach, uptake, usage patterns, impact on cancer risk factors and early detection practices (i.e., preliminary efficacy), perceived acceptability, appropriateness and usability and is not reported as part of the present paper.

## Data analysis

Interviews were audio-recorded in Microsoft Teams and transcribed verbatim. Interview facilitators (BM, KC, BV) reviewed transcripts for accuracy, corrected as needed and anonymised any identifiable comments. Transcript de-identification and cleaning was completed independently by all four facilitators and thematic analyses were conducted by BV. Transcripts were reviewed for clarity by interview facilitators with disagreements in perceived meaning resolved through discussion.

The cleaned and de-identified transcripts from each design stage were analysed (by BV) in NVivo V14 using an inductive thematic analysis approach as recommended by Braun & Clarke [54]. BM reviewed theme categorisation to improve reliability and validity of the data and the coding tree and its explanations were repeatedly discussed among the researchers (BV, KC, BM) until consensus was reached. The final codes were used to generate a list of themes that emerged in relation to the research objectives of each design stage. Summaries of themes were prepared for incorporation into subsequent workshops and complemented using data visualisation and user experience (UX) design tools such as personas and empathy maps. Personas and empathy maps are design thinking techniques used to explore the perspectives and experiences of consumers, ensuring that the end product meaningfully addresses their needs.. Personas involve the development of a series of fictional product users based on interview data, and are intended to anchor the development process around an understanding of *who* the product is designed for [55]. An empathy map is a collaborative tool that enables consumers to identify what product users "say, think, do, and feel", providing deeper insight into the needs and expectations of the intended user base [55]. Together, these tools aim to solidify consumer involvement in the design process to improve user experience of the end product [56,57].

To support prototype design, we applied a structured two-step approach based on established behaviour change constructs. First, we mapped the identified themes from stages one and two which captured frustrations and barriers to the COM-B (i.e., capability, opportunity, motivation-behaviour) framework domains and its respective sub-domains (i.e., physical and psychological capability, social and physical opportunity, automatic and reflective motivation) of the Behaviour Change Wheel [45]. The identification of specific intervention functions that could be used to address each theme was conducted in the second step and involved mapping the data from the third (i.e., ideation) stage to the corresponding evidence-based behaviour change techniques (BCTs).

## Results

### Study participants

A total of 110 out of 125 individuals who completed the expression of interest form were eligible to participate (Fig 1). Invitations to enrol as a panel member were sent to eligible individuals until a total of 30 consent forms were returned via reply email. Thirteen group-based and 17 individual workshops were held. Of the 30 consenting individuals, 22 were available to attend a workshop in Stage 1 (held in January to February 2024), 21 attended in Stage 2 (held in April to May 2024), 13 in Stage 3 (held in August to September 2024) and 17 in Stage 4 (held in November 2024). A total of 11 panel members attended a workshop at each of the four design stages, seven members attended three workshops, three attended two workshops, and two members attended only one of four workshops.

At the outset, two of the groups consisted of all male members, three groups consisted of all female members and a wide age range was represented in all five groups. The homogeneity observed between gender (all groups) and occupation (groups 1 and 3) was unintended but the research team did not intervene. Evidence suggests that while heterogenous focus groups facilitate new idea generation, homogeneity can drive connection, honesty, and openness [56,57], both of which are relevant processes in facilitating effective focus groups for this purpose. Table 2 contains further information on group composition, characteristics and reasons for participation.

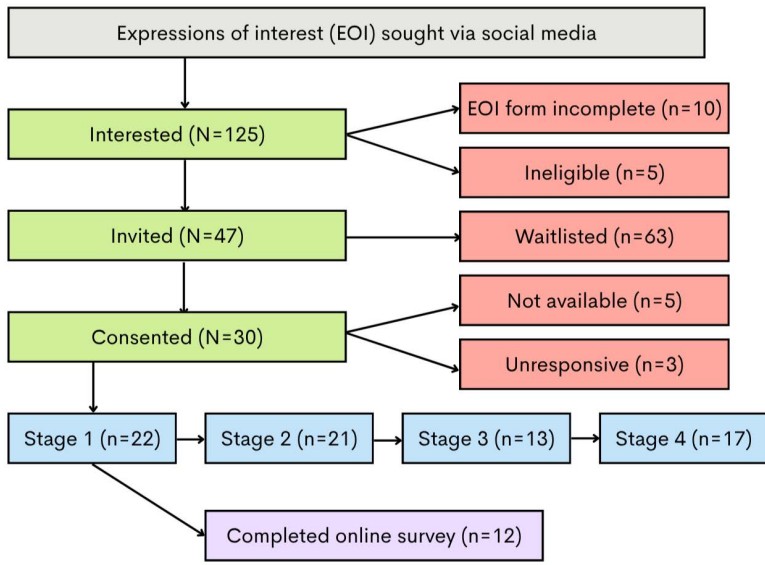

**Fig 1. Diagram illustrating the flow of panel members throughout the design process.**

**Table 2. Stage 1 panel member characteristics and group composition (N = 22)*.**

| | |
|---|---|
| **Gender N (%)** | |
| Female | 14 (64) |
| Male | 8 (36) |
| **Country of birth N (%)** | |
| Born in Australia | 19 (86) |
| Born overseas | 3 (14) |
| **Age (years)** | |
| Range | 23–73 |
| M (SD) | 43 (16) |
| **Geographic distribution N (%)** | |
| Southeast Queensland | 20 (91) |
| *Brisbane* | 12 |
| *Moreton Bay* | 3 |
| *Sunshine Coast* | 3 |
| *Darling Downs* | 1 |
| *Gold Coast* | 1 |
| Central Queensland | 2 (9) |
| *Bundaberg* | 1 |
| *Mackay* | 1 |
| **Group composition** | |
| Group 1 | 3 males, all with an IT background |
| Group 2 | 5 males, various occupations |
| Group 3 | 4 females from community services |
| Group 4 | 7 females, various occupations |
| Group 5 | 3 females, various occupations |
| **Main reason for participation N (%)** | |
| Having a relative/friend affected by cancer | 7 (32) |
| Personal history of cancer | 5 (23) |
| Desire to help/make a difference | 6 (27) |
| Interest in cancer research | 4 (18) |

*Note*. In later stages, Groups 1 and 2 (all-male groups) were merged into one. The three all-female groups were restructured to accommodate scheduling needs. Throughout all stages, gender composition remained consistent with exclusively male or female groups.

## Summary of panel perspectives: *Empathise* and *Define*

The key objectives of the *Empathise* and *Define* sessions were to empathise with end users and define the problems they are challenged with when seeking information and support services for cancer risk reduction and early detection. Based on the thematic analyses of data from these design stages, three main themes and a range of sub-themes representing prevalent frustrations and barriers related to cancer prevention and screening emerged (see Table 3).

## Summary of panel perspectives: *Ideate*

Then, in the *Ideate* stage, the aim was to shift from problems to solutions by asking the panel groups *how* the frustrations, barriers and needs identified during previous stages might be addressed. The brainstorming activities completed by the

**Table 3. Overview of common frustrations and barriers experienced by panel members.**

| Description | Example Statement |
|---|---|
| *Theme 1: Availability and knowledge of resources and support options* | |
| The groups consistently expressed dissatisfaction and confusion related to accessing and linking trustworthy pieces of information related to cancer prevention and early detection, including easy-to-comprehend and compelling research findings (i.e., in layman's terms, etc.). | *"There's no one place that's the official, credible source of all truth. That whole one stop shop platform that you're seeking this information from. Where do I go? Do I go to a government site? Do I go to a medical site?"* |
| Those ineligible for existing screening programmes (i.e., due to young age) expressed some frustration about the limitations of existing resources on self-checking for cancer symptoms (e.g., feeling for lumps or monitoring moles) and uncertainty about when and how to discuss concerns with healthcare providers. | *"I've definitely looked for ways to access screening services but there's not much available for young people and you have to wait around until you reach a certain age or meet the eligibility criteria, and it would be so cool if screening services were available earlier that, perhaps screen for other things that are still relevant to cancer and health risks."* |
| When encouraged to reflect on potential barriers to accessibility, panel members suggested there might be limited awareness of where to search for relevant resources and who to trust when it comes to digital health information, or issues with the practical application of generic information to individual circumstances and cancer risk more broadly. | *"Helping people access support is not always about solving problems for them, it's about empowering people to solve problems on their own and in line with their situation and needs. So, it's important to have a story behind the development of new initiatives/programmes and a clear intention or/ purpose and to do no harm."* |
| Amongst those eligible for existing screening programmes, concerns were shared about specific barriers to access (e.g., lack of local services, limited relevance/trust) encountered by those living in rural and remote areas and by First Nations people. | *"Seems like people want to be able to do more and be more engaged with their health, but they're coming across barriers in the system. Doesn't seem accessible for them. And they're becoming quite frustrated."* |
| *Theme 2: Information overload, accuracy and relevance.* | |
| This theme refers to people's ability to process, understand, and follow the health information currently available. For example, participants noted that existing information sources were either too generic, making it seem unachievable, irrelevant or unrelatable (i.e., cancer may not actively affect everybody in the present), or too dense, detailed, and complex, making it difficult to engage with. | *"I think sometimes the information's too general in terms of your own specific plan or your own specific needs. Or there's too much information. And you can get overloaded."* |
| Engaging others (e.g., peers, friends or family members) in cancer-related information is particularly challenging if they do not view themselves as at risk of cancer. | *"Until I was 40 or so, I'd had no major health problems. So, you know, I was feeling good. Thinking, I'm bulletproof. Everything's gonna be OK, and I'd never done any checkups or anything at all."* |
| There appears to be a breadth of misinformation about cancer risk and prevention online, potentially driving misconceptions as well as relativity biases (e.g., cancer as a disease of old age) and intentions to change. | *"One of the biggest things about misconceptions is that, as we all see with social media these days, it catches on like wildfire."* |
| *Theme 3: Time, money and motives* | |
| Existing solutions can be burdensome and/or expensive. With busy lifestyles and limited financial resources, many panel members admitted to struggling with the prioritisation of preventative health measures that offer no immediate, tangible benefits. | *"That's why it's so overwhelming, people are just so time poor. They're not even going to take themselves to the doctor. I know that I will only take myself to emergency. My children come first. Five doctor's appointments is a lot of money when you're not getting bulk billed (i.e., incurring out-of-pocket fees)."* |
| There is a lack of incentivisation regarding efforts towards cancer risk reduction. Panel members further discussed that those caught in the daily demands of work, with family or carer responsibilities, cancer prevention often becomes deprioritised against more pressing concerns with visible, short-term outcomes. | *"The biggest thing for me is that I'm just time poor and it's not a priority if it's about prevention of something that might happen in maybe 10 to 20, or 30 years, unless it's sun damage, which I'm very concerned about. You know, there are other things that I need to do in my immediate future right now. Like, I need to renew my driver's licence and pay my car registration. I need to find a place to live. There are all of these other immediate priorities, and it's just not a priority when there's such a big barrier with all the known complexities."* |

*Note.* Capability: A person's physical and psychological ability to perform the behaviour/s. Opportunity: External factors that facilitate or discourage the behaviour/s. Motivation: The desire to perform the behaviour/s at a specific moment.

panel groups generated a large volume of ideas and suggestions on what should be incorporated into the digital prototype. The data from this stage were categorised into five target themes. These are described in Table 4, with supporting quotations provided.

**Table 4. Overview of panel ideation and suggestions for the prototype.**

| Description | Example Statement |
|---|---|
| *Theme 1: Accessibility* | |
| A comprehensive, user-friendly platform that is widely promoted, ensuring easy access for all users through a memorable, cross-promotional approach (connecting users with existing Australia-based resources and services). | *"A system that's widely advertised so no one misses the opportunity to learn about the programmes that are in place. So, I don't know how that's done, but just needs to be out there for people to see to know about it".*<br>*"Links to access different organisations and help. So that you don't have to go out of the one platform and sort of open all different pages to view all different organisations.* |
| *Theme 2: Diversity* | |
| An inclusive design that caters to users with varying demographics, literacy levels, ages, and cultural backgrounds, by providing personalised, relevant content that resonates with different user groups. Snapshots of the latest research findings and learnings from the programme as such should be provided in a jargon-free format. | *"One of the things I thought was important was having a system that allows for various levels of IT experience. Because particularly if people are older, they may not…well, a lot of them can't use it very well. So, we need to allow for that. I'm not sure how that happens, but anyway".*<br>*"Design the platform in a way that is relevant to a broad and diverse cross section of the public so that the content and material published, or the platform can reach lots of different people".* |
| *Theme 3: Communication and connectedness* | |
| A robust communication system featuring interactive chat functions with both bot and human support, flexible notification options with user-controlled frequency, expert-moderated Q&A forums and feedback mechanisms to continuously improve user experience. In addition, there is a desire for people to connect with other users and peers, view and/or share testimonials and relatable stories, and to participate in local in-person events and current research projects or trials. | *"Somebody else mentioned some sort of push notification thing. I really like that idea as much as people hate push notifications. As long as you can turn it off and it's opt-in instead of opt-out, that's all good".*<br>*"I think it's a really valuable idea because it improves the website's function if there's a way of people giving that feedback. […] and maybe if there's some means of that feedback saying would you like to link to somebody that can give you more information or something?"*<br>*"Progress tracking links for more longer-term risk factor management with potential tie in with programmes such as quit smoking".*<br>*"Health trackers which could potentially tie into third party applications or tools to track the symptoms and potential risk factors. More so on the lifestyle side for, say, smoking and drinking".*<br>*"From a really positive user engagement perspective, allowing the app to interface with social media applications to share badge achievements particularly on Facebook and Instagram, where those kinds of badges are a nice fun way of really getting the awareness out there to friends and family and community".* |
| *Theme 4: Usability* | |
| An intuitive, user-friendly interface that supports assisted technologies, allows easy navigation (i.e., minimal time and clicks) and information management, provides a responsive design across different devices and includes features like voice commands and history tracking. | *"Voice commands. Available in different languages. That's probably not that unique and some websites have that".*<br>*"Ensure the platform's content can be understood by people from different cultural backgrounds. Have the ability to switch languages. So, on some websites you can toggle between having it presented in English or another language. You can basically click a button, and it can convert all the content to another language. I don't know how difficult that is to do that. That could be handy".*<br>*"Ensuring the platform [allows me to] use assisted technologies like screen read magnifiers, navigation switches and stuff like that. Like that's more for people who might have difficulty actually reading the content on screen".* |

**Table 4.** (Continued)

| Description | Example Statement |
|---|---|
| *Theme 5: Personal relevance* | |
| Users seek personalisation through tailored content based on their input (e.g., self-assessment forms or tools) and preferences (e.g., custom settings for reminders and notifications and a personal resource library). | *"Design the platform in a way that allows people to create a 'profile' that they can populate with as much or as little info as they personally desire".* <br> *"What came to my mind when I thought about an engaging app was having the opportunity to offer personalised content, based on a user profile, so particularly age, gender, a family history that can direct and steer and highlight certain risk factors and then following from that are tailor recommendations and reminders based on those particular risks from the user profile. So, for example, for females, regular breast screening check-ups".* <br> *Also tying into that is that you could probably also augment or chunk or split out the quiz journey across a few several modules in line with the achievement badges and that you could have a nice little short risk assessment quiz at the beginning of the journey, just nothing too overwhelming that can do some initial personalisation for their journey, and then as part of the achievement badges, they can be incentivised or enticed to further round out their risk assessments or risk improvement journeys by filling out further insights or information within the app as part of the achievement badge journey. So that's a neat way of chunking it so that can kind of get what you need up front without being overwhelming and at the same time give users a pathway to interact with the app as little or as much as they want in a fun and visually appealing way".* <br> *"And then potentially around the early detection support programme, I thought into that little bit further and thought about a section in the app for listing in the geolocation, all of the nearest screening areas that are geo optimised for the app user for them to know where their nearest screening area is based on their location".* |

Taken together, the panel members suggested the prototype should deliver actionable information by offering clear next steps with any recommendations made and provide a fully customised user experience based on full settings control (e.g., reminder types and frequency). Full settings control is a UX concept where users can have visibility, flexibility and mastery over their settings [58], allowing them to tailor the interface (e.g., content) to improve their satisfaction of the experience [59]. Short surveys (i.e., self-assessments) that tailor content further over time and the ability to filter and collate relevant information as needed were also suggested In addition, there was strong emphasis on the need for a social space where users could connect with one another.

In preparation for the *Prototype* stage, the themes that emerged from the ideation stage were mapped to the key intervention functions (i.e., education, persuasion, incentivisation, training, restriction, environmental restructuring and enablement) and the corresponding support strategies of the Behaviour Changes Wheel [45] (see S9 File).This visual representation confirmed a balanced distribution of intervention functions across all domains of the Behaviour Change Wheel.

## Digital design and prototype development

A comprehensive summary of insights derived from the *Empathise, Define,* and *Ideate* phases was presented to HG to determine requirements for the Lifestyle 6 prototype. To develop the visual representation of the prototype, HG initially

created a draft storyboard in Miro incorporating the community panel themes. The storyboard provided a central representation of the key insights shared by panel members their needs and preferences for the functionality of the prototype. It drew on user-suggested examples from existing apps developed to support health-related behaviour change. BM and GB reviewed this storyboard, providing community panel perspectives and programmatic expertise, respectively. After implementing their feedback, HG finalised the storyboard and developed the prototype using Articulate Storyline 360, configuring triggers and variables to simulate a mobile application experience with Cancer Council Queensland branding. Articulate Storyline 360 was selected as the authoring tool due to its ability to create mobile-friendly content including interactive elements and personalisation features. BM and JV independently audited the prototype to confirm it reflected the key themes as closely as possible. Further quality assurance checks were conducted (HG) before presentation to the community panel members, who, individually, were asked to review and appraise the prototype using the Think Aloud protocol.

The prototype (example screenshots shown in S10 File) was designed as a mobile-friendly application (i.e., for installation on smartphones and tablets) as this was favoured over a standard website by most panel members. To ensure that personal accounts and all user data are kept private, users were required to sign up with their email on first use; after which they were able to log in and out of their account at their convenience.

After registering an account, users were provided access to the start page of the programme, which asks them what they would like to get out of Lifestyle 6 with multiple priority areas available. These included: (1) understand and reduce my cancer risk, (2) understand how I can detect cancer early, (3) access cancer support, (4) connect with a community and (5) access evidence-based cancer information. The dashboard housed evidence-based information (e.g., facts sheets, research articles, videos, podcasts) and a range of support options (e.g., connect with other users by joining a community group, contact helpline, etc.), both of which align with the information provided by users and their selection of priority areas. Pre-filtered content was designed to be amended in the settings to further match individual preferences. Resources of interest were also able to be saved to a personal library for quick access.

Users are given the option to either accept the system-generated programme or amend it manually based on their preferences. They were also able to review and gradually adjust their goals in the programme builder section, define the actions they wish to take and/or join existing initiatives or challenges that align with their goal/s. The use of graded tasks and progress towards goals was instantly rewarded with encouraging feedback, whereas goal achievement was rewarded with respective badges. Within the goal setting section, users were able to view lists of current or future events and existing programmes and organisations that promote a healthier lifestyle, such as "10,000 steps" or "Dry July". However, participation in those programmes was entirely optional. Social comparison was facilitated through features that allow users to share their achievements on social media and across in-app community groups available in the prototype. The individual prototype testing workshops were used to seek user feedback (S11 File).

### Summary of panel perspectives: *Prototype*

During the *Prototype* phase, analyses revealed a notable emphasis on components linked to Psychological Capability and Physical Opportunity, with users consistently requesting more features that enhanced their understanding and improved the accessibility of information (e.g., easy to read text, larger icons and buttons, addition of buttons for further help and information). Suggestions related to customisation (i.e., tailoring to personal needs and preferences) also spanned multiple COM-B domains, but most strongly aligned with Automatic Motivation (e.g., provision of tangible rewards or incentives for progress and goal achievement). Social Opportunity elements were predominantly related to community engagement and support mechanisms (e.g., linkage to local support services, inclusion of real-life testimonials, opportunities for user interaction), with users seeking both educational and emotional support in their behaviour change journey. These insights will be incorporated into the final version of the prototype where feasible or implemented as part of future adaptations based on the evaluation of user data.

## Discussion

### Main findings

This study described the co-design of Lifestyle 6, a novel community-centric digital programme for cancer risk reduction and early detection support using established design thinking principles [36]. Thematic analyses of panel data revealed three distinct themes including: 1) having limited availability and knowledge of cancer-specific resources and support options, while simultaneously facing an overload of generic information; 2) concerns about the accuracy and relevance of available information and; 3) limited time, financial constraints, and insufficient motivation. This pattern appears consistently in the behaviour change literature for target behaviours such as physical activity, dietary changes, and screening behaviours impeded by similar barriers [60–63]. Our findings speak to the complexity of behaviour change [53] and highlight the need for a holistic digital behaviour change program with various entry points, and personalisation and support options throughout.

### Comparison to previous work

The analyses of panel data from the ideation workshops yielded five distinct themes, including accessibility, diversity, communication and connectedness, usability, and personal relevance. The suggested inclusions align with current best practices in digital health design [64] while highlighting several unique considerations for cancer prevention more specifically and collectively pointing toward a platform that balances comprehensive information with a custom experience, technological innovation with human connection, and scientific accuracy with accessible communication.

The need for jargon-free research messaging demonstrates the public's appetite for evidence-based information that is presented in accessible format. The panel members' specific request for expert-moderated forums may acknowledge potential issues of misinformation and conflict prevalent in unmoderated health communities. Moreover, the panel's recommendations for personalisation through self-assessment tools aligns with evidence that tailored interventions generally outperform generic approaches in changing health behaviours [65]. This is likely due to their impact on end-user persuasion as a key driver of behaviour change [56].

By suggesting cross-promotion with existing Australian resources or programs, panel members demonstrated awareness of the fragmented nature of health information systems and the need for integration rather than duplication. The emphasis on inclusive design addressing various demographics, literacy levels, and cultural backgrounds is particularly significant given the documented disparities in cancer outcomes across population groups [66]. It also aligns with longstanding calls for culturally responsive digital health solutions that move beyond mere translation to genuine cultural adaptation [67,68]. The desire for a comprehensive yet user-friendly platform reflects the tension identified in the literature between information completeness and cognitive overload [59]. The early involvement of professional designers and software developers is therefore crucial and helps in striking the right balance between the content presented and a high level of user experience [34].

### Implications for practice

Findings from this study highlight the use of tailored content to suit the individuals' needs, preferences, and stage of behavioural "readiness" for change as core user priorities [69]. However, this platform is intended to offer continual (rather than acute) support, and therefore a key responsibility of this platform will be to ensure that users are aware of the constraints in knowledge and advice that can be provided in the absence of a personal medical professional. As such, it will be critical to ensure that effective offboarding processes exist, connecting users with healthcare providers for individualised medical support. For example, partnership with community organisations may facilitate the development of self-referral pathways for individuals identified as "high risk" or in need of additional behaviour change support (such as those with high alcohol intake).

Furthermore, variations in risk perceptions related to cancer stood out as a barrier to behaviour change, however, the relationship between perceived risk and actual behaviour change is complex and often mediated by a vast spectrum of factors, including affect, self-efficacy, perceived behavioural control, social and environmental influences among others [70–72]. To address this complexity, several strategies warrant consideration. Health communication could be tailored and avoid messages perceived as threatening [72]. Personalised risk assessments that provide concrete, individualised feedback may prove more effective than generic risk information. Additionally, interventions could focus on building self-efficacy, as individuals must believe in their capacity to mitigate perceived risk [73]. Furthermore, addressing temporal discounting (i.e., the tendency to prioritise immediate rewards over long-term health benefits) remains essential [74,75]. This might involve framing behaviour change in terms of immediate positive outcomes rather than distant and abstract risk reduction. Social comparison and targeted campaigning, where individuals can benchmark their risk against peers, may also enhance the salience of risk perceptions and help correct some of the present misperceptions (i.e., cancer is a disease of the aged, no family history means no/low risk) [76].

## Strengths and limitations

The varied group compositions in this study enabled the collection of rich, multi-generational perspectives, while the gender-specific groups revealed distinct communication patterns. For example, male-only groups tended to provide concise, solution-focused contributions with frequent references to external examples and network experiences, whereas female groups typically offered more elaborate personal narratives and reflective insights, which is a commonly observed phenomenon [77]. The high proportion of panel members with personal cancer experiences or cancer within their immediate social networks may have contributed to elevated subject matter interest and heightened risk perceptions, that may not be typical of the broader population. Similarly, our decision to conduct the focus groups online may have enhanced our reach of participants outside of metropolitan areas, but it may also have resulted in selection bias, such that our sample may have exhibited higher levels of technological literacy than the general population. Offering in-person focus group attendance in future may combat this issue. Furthermore, despite our efforts to target diverse communities throughout our social media advertising campaign, people from low socioeconomic backgrounds, rural and remote communities, and Aboriginal and Torres Strait Islander populations were still under-represented. Future research should employ diverse recruitment strategies (such as collaboration with key community organisations) to more comprehensively reach these communities and enhance the breadth of insights gained. Overall, the virtual format for co-design workshops transformed rather than diminished collaboration, offering alternative pathways for engagement through features like chat functionality and breakout rooms that wouldn't have existed in traditional face-to-face workshops [78]. Finally, during data analysis all decisions about the inclusion and exclusion of information were clearly documented and discussed at each stage of the process, ensuring transparency and rigour.

## Conclusion

The insights reported above underpin the importance of striking the right balance between providing scientifically rigorous, yet palatable content and providing supportive community spaces in digital format. Novel programmes for cancer risk reduction and early detection would best consist of targeted, human-centred strategies that address current information gaps as well as the proliferation of online misinformation, ensuring individuals can easily locate and comprehend support options across various demographic groups. There is a need for tiered or personalised engagement strategies that account for varying levels of risk perception and different motivational drivers. Digital platforms should consist of intuitive interfaces that provide clear pathways to understanding the resources and support available and special attention must be given to the specific challenges and disparities faced by priority populations.

## Supporting information

**S1 File. Completed Consolidated Criteria for Reporting Qualitative Research (COREQ) checklist.**
(DOCX)

**S2 File. Project team.**
(DOCX)

**S3 File. Expression of interest form.**
(PDF)

**S4 File. Participant information sheet.**
(DOCX)

**S5 File. Privacy and confidentiality agreement.**
(PDF)

**S6 File. Ground rules for participation in online workshops.**
(DOCX)

**S7 File. Online feedback survey.**
(PDF)

**S8 File. Workshop structure and activities.**
(DOCX)

**S9 File. Relevant behaviour change techniques.**
(DOCX)

**S10 File. Intervention prototype screenshots.**
(DOCX)

**S11 File. Prototype feedback from panel members.**
(DOCX)

## Acknowledgments

We wish to sincerely thank the panel members for their time and valuable contributions, without whom this research would not have been possible. We also thank the Viertel Cancer Research Centre for their support of this research project.

## Author contributions

**Conceptualization:** Beatrice Murawski, Hannah Grennan, Grant Brown, Lee Woods, Sonja March, Belinda Goodwin.

**Data curation:** Beatrice Murawski, Katelyn Collins, Bianca Viljoen, Emma. Charlotte. Pearse.

**Formal analysis:** Beatrice Murawski, Katelyn Collins, Bianca Viljoen, Jazmin Vicario, Anna Stiller.

**Investigation:** Beatrice Murawski.

**Methodology:** Beatrice Murawski.

**Project administration:** Beatrice Murawski.

**Writing – original draft:** Beatrice Murawski.

**Writing – review & editing:** Beatrice Murawski, Katelyn Collins, Bianca Viljoen, Emma. Charlotte. Pearse, Jazmin Vicario, Anna Stiller, Hannah Grennan, Grant Brown, Lee Woods, Sonja March, Belinda Goodwin.

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
