## [Decision Letter · Decision Letter 0]

26 Nov 2025

Dear Dr. Collins,

Thank you for submitting your manuscript to PLOS ONE. After careful consideration, we feel that it has merit but does not fully meet PLOS ONE’s publication criteria as it currently stands. Therefore, we invite you to submit a revised version of the manuscript that addresses the points raised during the review process.

We look forward to receiving your revised manuscript.

Kind regards,

Dr Buna Bhandari

Academic Editor

PLOS ONE

**Journal Requirements:**

1. When submitting your revision, we need you to address these additional requirements. Please ensure that your manuscript meets PLOS ONE's style requirements, including those for file naming. The PLOS ONE style templates can be found at https://journals.plos.org/plosone/s/file?id=wjVg/PLOSOne_formatting_sample_main_body.pdf and https://journals.plos.org/plosone/s/file?id=ba62/PLOSOne_formatting_sample_title_authors_affiliations.pdf 2. Thank you for stating the following financial disclosure: This research was funded by Cancer Council Queensland.    Please state what role the funders took in the study.  If the funders had no role, please state: "The funders had no role in study design, data collection and analysis, decision to publish, or preparation of the manuscript." If this statement is not correct you must amend it as needed. Please include this amended Role of Funder statement in your cover letter; we will change the online submission form on your behalf. 3. We note that you have indicated that there are restrictions to data sharing for this study. For studies involving human research participant data or other sensitive data, we encourage authors to share de-identified or anonymized data. However, when data cannot be publicly shared for ethical reasons, we allow authors to make their data sets available upon request. For information on unacceptable data access restrictions, please see http://journals.plos.org/plosone/s/data-availability#loc-unacceptable-data-access-restrictions.  Before we proceed with your manuscript, please address the following prompts: a) If there are ethical or legal restrictions on sharing a de-identified data set, please explain them in detail (e.g., data contain potentially identifying or sensitive patient information, data are owned by a third-party organization, etc.) and who has imposed them (e.g., a Research Ethics Committee or Institutional Review Board, etc.). Please also provide contact information for a data access committee, ethics committee, or other institutional body to which data requests may be sent. b) If there are no restrictions, please upload the minimal anonymized data set necessary to replicate your study findings to a stable, public repository and provide us with the relevant URLs, DOIs, or accession numbers. Please see http://www.bmj.com/content/340/bmj.c181.long for guidelines on how to de-identify and prepare clinical data for publication. For a list of recommended repositories, please see https://journals.plos.org/plosone/s/recommended-repositories. You also have the option of uploading the data as Supporting Information files, but we would recommend depositing data directly to a data repository if possible. Please update your Data Availability statement in the submission form accordingly. 4. Please upload a new copy of Figure 2, as the detail is not clear. Please follow the link for more information:  https://journals.plos.org/plosone/s/figures 5. We note that Figures 2 and S4 in your submission contain copyrighted images. All PLOS content is published under the Creative Commons Attribution License (CC BY 4.0), which means that the manuscript, images, and Supporting Information files will be freely available online, and any third party is permitted to access, download, copy, distribute, and use these materials in any way, even commercially, with proper attribution. For more information, see our copyright guidelines: http://journals.plos.org/plosone/s/licenses-and-copyright. We require you to either present written permission from the copyright holder to publish these figures specifically under the CC BY 4.0 license, or remove the figures from your submission: a. You may seek permission from the original copyright holder of Figures 2 and S4 to publish the content specifically under the CC BY 4.0 license.  We recommend that you contact the original copyright holder with the Content Permission Form (http://journals.plos.org/plosone/s/file?id=7c09/content-permission-form.pdf) and the following text:“I request permission for the open-access journal PLOS ONE to publish XXX under the Creative Commons Attribution License (CCAL) CC BY 4.0 (http://creativecommons.org/licenses/by/4.0/). Please be aware that this license allows unrestricted use and distribution, even commercially, by third parties. Please reply and provide explicit written permission to publish XXX under a CC BY license and complete the attached form.” Please upload the completed Content Permission Form or other proof of granted permissions as an "Other" file with your submission.  In the figure caption of the copyrighted figure, please include the following text: “Reprinted from [ref] under a CC BY license, with permission from [name of publisher], original copyright [original copyright year].” b. If you are unable to obtain permission from the original copyright holder to publish these figures under the CC BY 4.0 license or if the copyright holder’s requirements are incompatible with the CC BY 4.0 license, please either i) remove the figure or ii) supply a replacement figure that complies with the CC BY 4.0 license. Please check copyright information on all replacement figures and update the figure caption with source information. If applicable, please specify in the figure caption text when a figure is similar but not identical to the original image and is therefore for illustrative purposes only. 6. We note that this data set consists of interview transcripts. Can you please confirm that all participants gave consent for interview transcript to be published? If they DID provide consent for these transcripts to be published, please also confirm that the transcripts do not contain any potentially identifying information (or let us know if the participants consented to having their personal details published and made publicly available). We consider the following details to be identifying information:- Names, nicknames, and initials- Age more specific than round numbers- GPS coordinates, physical addresses, IP addresses, email addresses- Information in small sample sizes (e.g. 40 students from X class in X year at X university)- Specific dates (e.g. visit dates, interview dates)- ID numbers Or, if the participants DID NOT provide consent for these transcripts to be published:- Provide a de-identified version of the data or excerpts of interview responses- Provide information regarding how these transcripts can be accessed by researchers who meet the criteria for access to confidential data, including:a) the grounds for restrictionb) the name of the ethics committee, Institutional Review Board, or third-party organization that is imposing sharing restrictions on the datac) a non-author, institutional point of contact that is able to field data access queries, in the interest of maintaining long-term data accessibility.d) Any relevant data set names, URLs, DOIs, etc. that an independent researcher would need in order to request your minimal data set. For further information on sharing data that contains sensitive participant information, please see: https://journals.plos.org/plosone/s/data-availability#loc-human-research-participant-data-and-other-sensitive-data If there are ethical, legal, or third-party restrictions upon your dataset, you must provide all of the following details (https://journals.plos.org/plosone/s/data-availability#loc-acceptable-data-access-restrictions):a) A complete description of the datasetb) The nature of the restrictions upon the data (ethical, legal, or owned by a third party) and the reasoning behind themc) The full name of the body imposing the restrictions upon your dataset (ethics committee, institution, data access committee, etc)d) If the data are owned by a third party, confirmation of whether the authors received any special privileges in accessing the data that other researchers would not havee) Direct, non-author contact information (preferably email) for the body imposing the restrictions upon the data, to which data access requests can be sent 7. If the reviewer comments include a recommendation to cite specific previously published works, please review and evaluate these publications to determine whether they are relevant and should be cited. There is no requirement to cite these works unless the editor has indicated otherwise. 

Reviewers' comments:

**Comments to the Author**

1. Is the manuscript technically sound, and do the data support the conclusions?

Reviewer #1: Yes

Reviewer #2: Yes

Reviewer #3: Partly

2. Has the statistical analysis been performed appropriately and rigorously?

Reviewer #1: N/A

Reviewer #2: N/A

Reviewer #3: Yes

3. Have the authors made all data underlying the findings in their manuscript fully available?

Reviewer #1: No

Reviewer #2: Yes

Reviewer #3: Yes

4. Is the manuscript presented in an intelligible fashion and written in standard English?

Reviewer #1: Yes

Reviewer #2: Yes

Reviewer #3: Yes

**Reviewer #1:** Thank you for the opportunity to review this interesting paper. This study is well designed and well explained in this manuscript.Thank you for the opportunity to review this interesting paper. This study is well designed and well explained in this manuscript.Thank you for the opportunity to review this interesting paper. This study is well designed and well explained in this manuscript.Thank you for the opportunity to review this interesting paper. This study is well designed and well explained in this manuscript.

Introduction:

The need for an intervention with the capacity to address the multiple risk-factors across a number of cancer types is well motivated, as are the advantages of mHealth solutions.

The adoption of the Co-Design approach gives voice to likely end users and is well supported by the referenced literature.

Methods:

The application of Design Thinking is well motivated and appropriate. The referenced Learning Health Systems Framework and Behaviour Change Wheel give structure and the opportunity to translate findings into practical applications, important when the overall aim is to design a useful tool. The manuscript meets the COREQ criteria, as detailed in the Supplementary File.

The adoption of videoconferencing for engagement with panel members is likely to have aided recruitment and engagement. Was the adoption of this engagement strategy motivated by a review of the success of online engagement in other studies or was this for convenience? A comment on this choice could be added.

Was there any PPI involvement in the design of the engagement process?

How the participants were allocated to the initial 5 groups could do with further explanation, particularly how homogeneity was defined. Was it intended that Group 1 would all be IT professionals? Same for the group comprised of community workers. Why did each group consist of females or males only? Explain why homogeneity might be appropriate.

Timelines for the process should be explained. How much time elapsed between workshops?

Participants and recruitment: The limitations section notes that greater representation from low socioeconomic backgrounds, rural and remote communities, and other priority populations would have further enhanced the breadth of insights gained. The online format could have helped with this. Other studies engaged with representative organizations/NGOs in order to reach these seldom heard voices. What consideration was given to reaching these populations?

Additional information could be added to the Supplementary Files:

• the participant information statement, the privacy and confidentiality agreement and the ground rules for participation in the online workshops.

• The online survey completed by participants.

Results & Discussion

Well presented. Findings are interesting, clearly explained and adequately discussed.

On completion of the engagement, were the panel members given an opportunity to provide feedback on the engagement process?

Figure 2 – I struggled to read this image.

Overall, this study/manuscript is an interesting read and makes a valuable contribution to the literature.

**Reviewer #2:** Dear Manuscript Authors, Dear Manuscript Authors, Dear Manuscript Authors, Dear Manuscript Authors,

Thank you for writing a paper aimed at enhancing knowledge about incorporating community feedback to develop a mobile health app product. Please consider the following comments to provide more clarity in your writing, as well as expand a bit more on the practice implications and inclusivity/diversity of your sample population based on study eligibility and recruitment. Best wishes!

Introduction Section

Line 53: I suggest clarifying that it’s solid tumour cancers for prevention via healthy lifestyle.

Lines 57-58: Please carefully re-write this sentence to make evident certain cancer diseases (a small portion) have evidence-based established guidelines for early detection cancer screening. This could be a supporting reason for why it is critical to consider interventions that can address modifiable behaviors cancer risk factors due to limitations with cancer screening for other cancers.

If you can cite research knowledge relevant to populations in Australia or high-income countries (as a whole), briefly highlight existing evidence related to modifiable cancer risk factors. For example, evidence similar to the paper written by Islami et al. (2024) Proportion and number of cancer cases and deaths attributable to potentially modifiable risk factors in the United States, 2019. CA: A Cancer Journal for Clinicians.

Participants and Recruitment

Lines 132 – 134: There is mention of “panel members.” Were all participants a panel member? Re-word to make it clear that all or some participants’ role was to be engaged as panel member.

Data Collection

Line 151: Make it clear that the individual workshops or all the workshops were online.

Discussion

Regarding implications for practice, have you considered the feasibility of maintaining an educational/information dissemination product that is accurate, up-to-date, clearly evidence-based, and engaging for the user? How would the users be encouraged to communicate with GP (or other medical professional) to discuss their unique health situation?

Strengths and limitations

The web-based participation is convenient and practical since it allows for human subjects to participate from their personnel space. Although have you considered that the web-based participation requirement may have limited certain types of participants from being a valuable informant? Do you think your data reflects well the younger adult population perspectives and experiences (18 – 30 years old)? Future direction could involve in-person study participation and improve recruitment of young adult populations for the next phases of testing the product.

**Reviewer #3:** This article describes the co-design of Lifestyle6, a digital health program to reduce cancer risk and support early detection through multiple behavior changes. Using design thinking, the authors engaged multiple community members in iterative workshops to identify barriers, define needs, and prototype a mobile-friendly solution. The study addresses critical gaps in cancer prevention by integrating user-centered design to improve engagement and personalization—key challenges in digital health interventions. Its importance lies in advancing scalable, evidence-based strategies for cancer risk reduction and screening support, which could significantly impact public health outcomes.This article describes the co-design of Lifestyle6, a digital health program to reduce cancer risk and support early detection through multiple behavior changes. Using design thinking, the authors engaged multiple community members in iterative workshops to identify barriers, define needs, and prototype a mobile-friendly solution. The study addresses critical gaps in cancer prevention by integrating user-centered design to improve engagement and personalization—key challenges in digital health interventions. Its importance lies in advancing scalable, evidence-based strategies for cancer risk reduction and screening support, which could significantly impact public health outcomes.This article describes the co-design of Lifestyle6, a digital health program to reduce cancer risk and support early detection through multiple behavior changes. Using design thinking, the authors engaged multiple community members in iterative workshops to identify barriers, define needs, and prototype a mobile-friendly solution. The study addresses critical gaps in cancer prevention by integrating user-centered design to improve engagement and personalization—key challenges in digital health interventions. Its importance lies in advancing scalable, evidence-based strategies for cancer risk reduction and screening support, which could significantly impact public health outcomes.This article describes the co-design of Lifestyle6, a digital health program to reduce cancer risk and support early detection through multiple behavior changes. Using design thinking, the authors engaged multiple community members in iterative workshops to identify barriers, define needs, and prototype a mobile-friendly solution. The study addresses critical gaps in cancer prevention by integrating user-centered design to improve engagement and personalization—key challenges in digital health interventions. Its importance lies in advancing scalable, evidence-based strategies for cancer risk reduction and screening support, which could significantly impact public health outcomes.

My overall feedback is that when I read the paper, I remained confused about what was being developed and the process of development. It wasn’t until I saw the supplements showing the interface that it started to make sense to me, and the supplement that described the workshop structure and activities by stage, and the supplement with the Prototype screenshots. I think my feedback is that the manuscript needs a little more structure, and parallel writing to help the reader understand what is being done and why at each step. Naming or numbering each step consistently might help, or perhaps subheadings would be useful.

My second high level feedback is about the interdisciplinary nature of this paper, which is a strength, but the manuscript would be improved by more clearly defining design-thinking and user experience type constructs mre clearly from the beginning of the manuscript. Overall, there was a lot of describing of “behavior change theories”, largely drawn from health promotion, social psych, etc. type fields of behavioral and social sciences without an equivalent set of terms or elements drawn from design fields. This is a missed opportunity.

Nevertheless, this provides something useful and novel and I think Plos One readers will be interested. The process is scalable and with some suggested editing of the manuscript, I think it makes a good contribution because it provides rationale and a process for co-design. The focus on multiple behaviors is also unique and critical for effective cancer prevention and control.

Below are some minor questions and suggestions

Line 188: Describe “padlet and chat posts” . As a reader I don’t know what these are.

191- Define COREQ guidelines and at least spell out.

192- Clarify who and how “appraise the standard of evidence” was done. This seems like a very important step and there’s not enough details for me to have confidence in this process.

201- personas and empathy maps need to be explained somewhere for those who aren’t experts in this topic. I wonder if the design thinking concepts like this could be added to one of the tables in the supplement that lists all the “behavior change theories” that are elements of theories.

Grammatical confusion in table 3, last row under theme 1 sentence starting “amongst those eligible…”

Line 257- wat is “full settings control”?

Line 256-6: The phrase “This step was completed to facilitate reporting” confuses me. And how is this strong rationale for doing this work? What reporting is being referred to here, and what does that have to do with intervention design and development?

Line 300- This was confusing to me saying things “can be used” made me wonder- is this how the prototype currently works, or a future adaptation that is possible given the prototype design. .

Fig 2 confusing especially part in top right with game of Life and random images.

Minor point- most of the supplements are mislabeled in the online system file name, making it a little confusing.

Another minor point. Several times it is mentioned that intervention components are matched behavioral change theories. This is a little bit of a stretch because few theories are listed, it’s specific theoretically-driven constructs that are being discussed.

.

Reviewer #1: No

Reviewer #2: No

Reviewer #3: No

---

## [Author Response · Author response to Decision Letter 1]

23 Feb 2026

Please see attached file, "Response to reviewers_22122025" which includes a point by point response to reviewer feedback.

---

## [Decision Letter · Decision Letter 1]

1 Apr 2026

Co-design of Lifestyle6, a digital tool targeting multiple health behaviour changes for cancer risk reduction and early detection support

PONE-D-25-26923R1

Dear Dr. Collins,

We’re pleased to inform you that your manuscript has been judged scientifically suitable for publication and will be formally accepted for publication once it meets all outstanding technical requirements.

Kind regards,

Dr Buna Bhandari

Academic Editor

PLOS One

Additional Editor Comments (optional):

Reviewers' comments:

Reviewer's Responses to Questions

**Comments to the Author**

Reviewer #1: All comments have been addressed

Reviewer #2: All comments have been addressed

2. Is the manuscript technically sound, and do the data support the conclusions?

Reviewer #1: Yes

Reviewer #2: Yes

3. Has the statistical analysis been performed appropriately and rigorously?

Reviewer #1: N/A

Reviewer #2: Yes

4. Have the authors made all data underlying the findings in their manuscript fully available?

Reviewer #1: Yes

Reviewer #2: Yes

5. Is the manuscript presented in an intelligible fashion and written in standard English?

Reviewer #1: Yes

Reviewer #2: Yes

Reviewer #1: The authors have improved this manuscript based on the comments raised by all reviewers and I am happy to recommend publication.

Reviewer #2: 227: remove additional punctuation

You've produced a superb paper with your revisions. Best wishes for continued work in this area.

.

Reviewer #1: No

Reviewer #2: No

---

## [Editor Report · Acceptance letter]

PONE-D-25-26923R1

PLOS One

Dear Dr. Collins,

I'm pleased to inform you that your manuscript has been deemed suitable for publication in PLOS One. Congratulations! Your manuscript is now being handed over to our production team.

Kind regards,

on behalf of

Dr. Buna Bhandari

Academic Editor

PLOS One